# Toxicity Assessment of an Anti-Cancer Drug of p-Toluene Sulfonamide in Zebrafish Larvae Based on Cardiovascular and Locomotion Activities

**DOI:** 10.3390/biom12081103

**Published:** 2022-08-10

**Authors:** Andrew Yau Wah Young, Gilbert Audira, Ferry Saputra, Honeymae C. Alos, Charlaine A. Aventurado, Yu-Heng Lai, Ross D. Vasquez, Chung-Der Hsiao, Chih-Hsin Hung

**Affiliations:** 1Institute of Biotechnology and Chemical Engineering, I-Shou University, Kaohsiung 84001, Taiwan; 2Department of Bioscience Technology, Chung Yuan Christian University, Chung-Li 320314, Taiwan; 3Department of Chemistry, Chung Yuan Christian University, Chung-Li 320314, Taiwan; 4The Graduate School, University of Santo Tomas, Manila 1008, Philippines; 5Department of Chemistry, Chinese Culture University, Taipei 11114, Taiwan; 6Department of Pharmacy, Research Center for Natural and Applied Sciences, University of Santo Tomas, Manila 1008, Philippines; 7Research Center for Aquatic Toxicology and Pharmacology, Chung Yuan Christian University, Chung-Li 320314, Taiwan

**Keywords:** p-TSA, zebrafish, larva, neurotoxicity, cardiotoxicity

## Abstract

p-Toluene sulfonamide (p-TSA), a small molecular drug with antineoplastic activity is widely gaining interest from researchers because of its pharmacological activities. In this study, we explored the potential cardio and neural toxicity of p-TSA in sublethal concentrations by using zebrafish as an in vivo animal model. Based on the acute toxicity assay, the 96hr LC_50_ was estimated as 204.3 ppm, suggesting the overall toxicity of p-TSA is relatively low in zebrafish larvae. For the cardiotoxicity test, we found that p-TSA caused only a minor alteration in treated larvae after no overall significant alterations were observed in cardiac rhythm and cardiac physiology parameters, as supported by the results from expression level measurements of several cardiac development marker genes. On the other hand, we found that acute p-TSA exposure significantly increased the larval locomotion activity during the photomotor test while prolonged exposure (4 days) reduced the locomotor startle reflex activities in zebrafish. In addition, a higher respiratory rate and blood flow velocity was also observed in the acutely treated fish groups compared to the untreated group. Finally, by molecular docking, we found that p-TSA has a moderate binding affinity to skeletal muscle myosin II subfragment 1 (S1), ATPase activity, actin- and Ca^2+^-stimulated myosin S1 ATPase, and v-type proton ATPase. These binding interactions between p-TSA and proteins offer insights into the potential molecular mechanism of action of p-TSA on observed altered responses toward photo and vibration stimuli and minor altered vascular performance in the zebrafish larvae.

## 1. Introduction

The sulfonamides are a family of antibiotics comprising sulfadimethoxine, sulfamethoxazole, and trimethoprim observed in surface waters worldwide [1]. Tropical Asia has observed sulfonamide as a common contaminant in wastewater sewage, mainly from pig farms [2]. The research group of Chen and Zhou et al., 2014, and Zhang et al., 2015b, reported 7890 tons of sulfonamide consumption in 2013. Thus, sulfonamides were found at an elevated concentration in rivers since sulfonamides are not efficiently removed by wastewater treatment plants [3,4]. Sulfonamides are used in various synthesis applications in high molecular weight substances and pharmaceuticals [5]. The composition of sulfonamides is known as synthons for the production of different biologically active compounds such as artificial antifolic agents [6]. Sulfa drugs are used as antibiotics [7,8]; they hinder the p-aminobenzoic acid during the biosynthesis of tetrahydrofolic acid required for bacterial metabolism [9]. Some sulfonamide derivatives have shown commendable anti-cancer activity in vitro and in vivo. Aromatic sulfonamides have been reported to interfere with the growth of tumor cells [10,11,12,13,14,15]. Similarly, established uses of sulfonamides are antifungal [16], antiviral [17], anti-inflammatory [18], hypoglycaemic [19], and protease inhibitor agents [20].

P-toluene sulfonamide (p-TSA) with molecular formula C_7_H_9_NO_2_ and IUPAC name 4-methyl benzenesulfonamide is a prominent example of an organic compound of the sulfonamide group and has gathered the interest of researchers widely because of its notable pharmacological activities [5]. It is used as an intermediate in the organic synthesis of dyes, resins, pesticides, thermoset plastic, and organic target compounds [21,22,23,24]. p-TSA is partially compatible with cellulose nitrate and cellulose acetate but incompatible with vinyl chloride copolymer and polyvinyl chloride. p-TSA is stable in acidic, alkaline, and neutral conditions. It is a non-volatile chemical that appears as a crystalline powder or white flakes in the solid form [25]. p-TSA has been approved by the US Food and Drug Administration (FDA) for application into adhesives as food packaging materials. A few years ago, p-TSA was recognized as a new anti-cancer agent with superior lipophilic properties [26]. Moreover, it is used as an anti-cancer drug for various cancers, namely tongue squamous cell carcinoma, non-small cell lung cancer, and hepatocellular carcinoma. p-TSA has also shown competent anti-tumor activity in clinical trials via local injection [26,27,28]. p-TSA is lipophilic and can cross and distribute efficiently within a tumor [25].

There is no doubt that p-TSA can offer many benefits and uses; however, toxicity studies of p-TSA are minimal. A study has reported p-TSA to be minimally toxic to algae but not to daphnids and fish, suggesting that environmental risk is very minute [29]. Recently, the toxic effects of sulfonamide have been reported in zebrafish on the parameters of behavior and reproduction, sulfonamide comprising sulfadimidine, sulfamethoxazole, and sulfadiazine at a concentration range of 1 µg/L to 10 mg/L caused a decrease in swimming activity and an increase in heart rate of zebrafish [30]. Similarly, an increase in glutathione S-transferase (GST) and malondialdehyde (MDA) activity was observed in zebrafish on exposure to sulfonamide, suggesting GST and MDA are potential biomarkers of toxicity in fishes [31,32,33]. Therefore, based on these observations, more studies are needed to analyze its toxic effects in animal models. Here, we conducted a novel study to analyze the acute toxic effects of p-TSA in zebrafish larvae using parameters such as cardiac physiology, cardiac rhythm, photomotor response, startle reflex, and respiratory rate in addition to molecular docking of p-TSA to its target proteins and measurement of the expression level of several cardiac development marker genes.

## 2. Material and Methods

### 2.1. Zebrafish Maintenance

Adult zebrafish AB strains used for breeding were obtained from Taiwan Zebrafish Stock Center at Academia Sinica (TZCAS) and have already been bred for several generations. The fishes were kept in a continuous aerated water system at 26 ± 1 °C with 14/10 h of light/dark cycle. The rearing condition was based on the previous study [34]. During the breeding process, one female fish and two male fishes were placed in a breeding chamber for the mating process. Afterward, the fertilized eggs were collected and incubated in distilled water with 0.0001% methylene blue to act as a fungicide at 28 ± 1 °C before the exposure. In the present study, we only used healthy embryos and larvae without any infections. The experiment was conducted according to the guidelines for the care and use of Laboratory Animals by Chung Yuan Christian University (CYCU) and approved by the Animal Ethics Committee of CYCU.

### 2.2. Acute Toxicity Test

After assessing the quality of the embryos, embryonic toxicity tests were performed. Healthy embryos were distributed into a 5.5 cm diameter-Petri dish with 10 embryos per plate. An acute exposure regime of 96 h was used, from 24 h to 120 h post-fertilization (hpf), including the major organ development stages. In determining the 96h-LC_50_ of p-TSA, the embryos were exposed to 0.01, 0.1, 1, 10, 100, 150, 200, 250, 500, and 100 ppm of p-TSA in the water with the same composition as the one used for the incubation of the eggs and kept at 25 ± 0.5 °C. The volume of liquid was 20 mL in each Petri dish. The number of dead embryos in individual concentrations was recorded during the test. Triplicate trials involved ten embryos per treatment group (a total of 30 embryos) were applied in this test. The embryonic toxicity test was based on a previous study [35].

### 2.3. Cardiotoxicity Assessment

At 2 days post fertilization (dpf), healthy zebrafish larvae were selected and incubated with 10, 50, and 100 ppm (*w*/*v*) of p-TSA for 24 h. These concentrations were sub-lethal concentrations that were based on the results obtained from the acute toxicity test. The assessment for cardiac performance was done following the protocol by Saputra et al. 2021 [36]. In short, the zebrafish larvae were observed under an inverted microscope (ICX, Sunny Optical Technology, Zhejiang, China) mounted with high performance coupled charged device (CCD) (AZ Instrument, Taichung City, Taiwan) camera capable of recording video at 200 frames per second (fps). Hoffmann Lens with 40× and LPlan Lens with 10× magnification were used to observe the cardiac chamber and cardiac rhythm, respectively. In reducing the movement of zebrafish during recording, 3% methylcellulose was used as the mounting solution. The recording was done for 10 s for each fish with ten biological replications for each group. The video was then processed with the software ImageJ to calculate the cardiac performance parameter. The Time-series analyzer plug-in (Available online: https://imagej.nih.gov/ij/plugins/time-series.html (accessed on 22 April 2022)) in ImageJ was used to analyze the cardiac rhythm according to the previous study [36,37]. The experiment was done in three replications.

### 2.4. Vascular Performance Assessment

The vascular performance assessment was done by analyzing the blood flow velocity in the dorsal aorta. Vascular performance assessment was done in the same setting as the cardiac physiology analysis. Hoffmann lens with 40× magnification was used to observe the dorsal aorta region of zebrafish larvae. The recording was done for 10 s with 200 fps. Trackmate Plugin in ImageJ was used to calculate the blood flow velocity according to the method described by Santoso et al. [38,39]. The experiment consisted of ten biological replications for each group and three technical replications.

### 2.5. Photomotor Response and Morphology Assessment

In evaluating p-TSA neurotoxicity, the behavioral activity of exposed larvae was assessed. First, the photomotor response test was carried out, which relies on the integrity of both eye and locomotor/skeletal system development. After 24 h of incubation, 120 hpf zebrafish larvae were placed individually into each well of a 48-well transparent plastic plate and placed into ZebraBox (Viewpoint, Civrieux, France). Later, each group was subjected to the light and dark challenge test with a total of 80 min of recording, consisting of four light cycles and four dark cycles with 10 min duration for each cycle. Before the test, for the first 30 min, the light was off as a pretest adaptation period. ZebraBox recording apparatus recorded the behavior for the next 80 min. Their swimming patterns and a movement distance per minute were analyzed with VideoTrack software (both from Viewpoint). In addition, larval photomotor response (LPMR) was also calculated to measure the swimming responses of the larvae to a sudden change in light condition by calculating the change in mean distance traveled between the last minute of an initial photoperiod and the first minute of the following period. All settings used in the protocol were based on previous publications [40,41]. This experiment was done in three replicates with a total number of ~134 larvae for each group. In addition, by using the same exposure method, the morphology of another batch of larvae was assessed. Here, two physiological endpoints, which were body and eye axis lengths, were measured. This analysis was done in three replicates with a total number of 90 larvae for each group.

### 2.6. Vibrational Startle Response Assessment

Similar to the photomotor response test, a vibrational startle response assay was also conducted to evaluate p-TSA neurotoxicity, especially in the zebrafish larval behaviors, by measuring their activity as a response to repetitive vibrational stimuli generated by a tapping device. Here, we tested the startle response of zebrafish larvae that survived the photomotor response test after 96 h of p-TSA exposure. At 168 hpf, all living larvae were placed in ZebraBox and were acclimated for 30 min. Afterward, the test was started with the first 5 s without any stimulus, followed by 20 s of tapping stimulus at the highest intensity with a 1-s interstimulus interval. The distance traveled by each larva every one second was recorded. The current procedure was based on the prior study and carried out in three replicates with a total sample size of 133, 135, 121, and 104 for control, 10, 50, and 100 ppm groups, respectively [42].

### 2.7. Respiratory Rate Assessment

After 24 h of incubation with p-TSA, the oxygen consumption rate assay was carried out to evaluate the effect of this compound on the respiratory rate of 96 hpf zebrafish larvae by Microplate system by Loligo Systems (Loligo Systems, Viborg, Denmark). Each tested larva was transferred to a 24-well plate with 80µL of the solution above a Sensor Dish Reader (SDR) with one empty well as a blank. Afterward, the oxygen consumption every 30 s for 60 min was measured by MicroResp™ software version 1.0.4 (Loligo Systems, Viborg, Denmark). The current method is based on previous publications [43,44].

### 2.8. Molecular Docking of p-TSA to Its Target Proteins

The proteins/receptors for docking were selected based on their impact or relationship to development, cardiac and vascular contraction, locomotion, and behavioral reflexes on zebrafish larvae. The structure of the p-TSA was accessed from the PubChem (https://pubchem.ncbi.nlm.nih.gov/ (accessed on 31 May 2022)) while the zebrafish protein receptors were obtained from the AlphaFold protein structure database (https://alphafold.ebi.ac.uk/ (accessed on 31 May 2022)). The AutoDock Vina wizard v1.1.2 in the Python Prescription (PyRx) virtual screening software (https://pyrx.sourceforge.io (accessed on 31 May 2022)) was used for reverse docking [45]. PyRx’s Open Babel was utilized to prepare structures of the ligands, while PyRx’s AutoDock Tool was used to process and prepare the protein of interest. Using the AutoDock Tools in PyRx, a PDBQT file of the ligand and receptor required for docking was prepared. A receptor grid was generated around the protein’s target region to allow p-TSA to dock in a specified binding location. The grid box size utilized was presented in Table 1. The p-TSA was docked using PyRx’s AutoDock Vina wizard using the Assisted Model Building with Energy Refinement (AMBER) force field as the scoring parameter. The interactions were graphically visualized and analyzed using the Discovery Studio (DS) 2021 Client (https://www.3dsbiovia.com (accessed on 31 May 2022)).

### 2.9. Gene Expression Analysis

Based on the alterations in locomotion activity and vascular performances caused by p-TSA, the expression level of several genes that are related to the induction of cardiac development was measured. The total RNA of zebrafish larvae at 128 hpf was isolated after 1-day exposure to p-TSA by using RNAzol reagent (Molecular Research Center, Inc., Cincinnati, OH, USA) according to the manufacturer’s protocol. After the RNA’s quality was assessed y UV spectrophotometry, the total RNA was reversed to cDNA by using SuperSScript II transcriptase (18064-014, Thermo Fisher Scientific, Waltham, MA, USA) according to the manufacturer’s protocol. Each primer’s sequence is listed in Table A1 and was designed by Integrated DNA Technologist (Coralville, IA, USA). Quantitative real-time PCR (qRT-PCR) amplifications were performed with the following parameters: denaturation for 20 s at 95 °C, followed by 30 cycles at 95 °C for 3 s and 58 °C for 30 s that the fluorescent signal was measured at the annealing/extension step. Beta-actin (β-actin) was used as a control and qRT-PCR amplifications were performed on a Real-time PCR system StepOne System (Thermo Fisher Scientific, MA, USA). Later, melt curve analysis was performed to validate the specificity of the PCR products. Each mRNA level was expressed according to its ratio to β-actin mRNA. This experiment was conducted in three biological replicates with 70 zebrafish larvae for each replicate and the relative quantification of gene expression among the groups was calculated by using the 2^−ΔΔCt^ method [46].

### 2.10. Statistical Analyses

Before statistical analyses were conducted to compare the treated groups with the control group, the data normality of all groups was tested to determine the normality of data distribution. For normally distributed data, one-way ANOVA was used to calculate the statistical differences between control and treated groups. If the data distribution was not normal, the non-parametric Kruskal–Wallis followed with Dunn’s multiple comparisons test was used since this analysis does not require a normal distribution assumption [47]. In addition, mixed-model two-way ANOVA continued with Sidak’s multiple comparisons tests was also used in some results to determine how the behaviors are affected by two factors. All data with normal distribution are expressed as the mean with standard error mean (SEM). In contrast, other data without normal distribution are expressed as the median with an interquartile range since, generally, it is used to describe data in a skewed distribution, including behavior data [48,49]. The statistic differences between the control and treated groups are indicated either with “*” (*p* < 0.05), “**” (*p* < 0.01), “***” (*p* < 0.001), or “****” (*p* < 0.0001). The statistical analyses and graph plotting were conducted with GraphPad Prism (GraphPad Software version 8 Inc., La Jolla, CA, USA). All of the analyses were conducted by blind-trained analysts.

## 3. Results

### 3.1. Acute Toxicity Test of p-TSA

The toxicity tests showed that the 96 h LC_50_ value of p-toluene sulfonamide for zebrafish larvae was around 204.3 ppm with a 95% confidence interval (CI) in the range of 189.5 to 215.7 (Figure 1). Comparing the results of the experiment with the acute toxicity rating scale provided by the U.S. Fish and Wildlife Service (USFWS) and the United States Environmental Protection Agency (USEPA) revealed that p-TSA is categorized as a practically nontoxic compound to aquatic organisms, especially fish [50,51,52].

### 3.2. Cardiac Performance Assessment of p-TSA in Zebrafish

A cardiac performance assessment was done by analyzing the cardiac physiology and rhythm parameters. Several parameters including stroke volume, cardiac output, heart rate, ejection fraction, and shortening fraction were calculated to assess cardiac physiology. Moreover, the heart rate variability and atrium-ventricle beat interval were calculated to assess the cardiac rhythm. Based on LC_50_ data, the sub-lethal doses of 10, 50, and 100 ppm were further analyzed to check the potential cardiotoxicity in zebrafish larvae. This study observed a significantly higher stroke volume in zebrafish larvae after incubation in 100 ppm of p-TSA. Furthermore, significant decrement was also observed in the heart rate of both chambers after 10 and 100 ppm of p-TSA (Figure 2A,B,E). However, the absence of significant difference in cardiac output, ejection fraction, and shortening fraction of every group tested suggests that p-TSA only causes minor alteration in zebrafish cardiac physiology (Figure 2C,D,F). Following the cardiac physiology result, no significant change in every cardiac rhythm parameter was also observed in zebrafish exposed to p-TSA (Figure 3). These results suggest that p-TSA did not cause any significant cardiac performance alteration up to 100 ppm concentration.

### 3.3. Vascular Performance Assessment of p-TSA in Zebrafish

A vascular performance assessment was done by analyzing the maximum and average blood flow velocity of zebrafish larvae after p-TSA exposure. Contrary to the cardiac performance data, a significant increase in maximum blood flow velocity was observed in 50 and 100 ppm concentrations (Figure 4). However, a significant increase in average blood flow velocity was only observed at 50 ppm concentration, which means that 50 ppm might be the optimal dose to cause a significant increase in zebrafish blood flow velocity.

### 3.4. Photomotor Response and Morphology Assessment of p-TSA in Zebrafish

Since it belongs to the sulfonamides group, it is intriguing to evaluate the neurotoxicity of p-TSA in zebrafish larvae because several sulfonamides were reported to induce apparent effects on spontaneous swimming activity in zebrafish [30]. Here, the zebrafish larval responses toward light changes after p-TSA exposure were evaluated using the PMR test. The treated fishes showed a statistically similar response pattern toward the changes in both light and dark cycles (Figure 5A and Appendix A); however, the magnitudes of activity of these treated groups differed from the control group. This phenomenon was indicated by the statistically higher levels of distance traveled by higher concentration groups, which were 50 and 100 ppm, than the untreated group (Figure 5B,C). However, the lowest dose group exhibited a comparably similar activity level to the control group during the test (Figure 5B,C). These results suggest that acute exposure to p-TSA in high doses induced hyperactivity-like behaviors in zebrafish larvae. Interestingly, despite their behavior abnormalities, the treated larvae showed a comparably normal morphology to the untreated group, shown by the similar values of body and eyes axis length between these groups (Figure A2).

### 3.5. Startle Reflex Assessment of p-TSA in Zebrafish

Based on the photomotor test results in the previous section, we hypothesized that p-TSA could also impair zebrafish larval response toward vibration stimuli. Therefore, a vibrational startle response assay (VSRA) was carried out to test this hypothesis. In this test, the total distance traveled by each larva was measured before and during exposure to repetitive vibrational stimuli generated by the tapping device. Here, the fishes from the photomotor tests were used since, based on a prior study, VSRA was conducted on 8 dpf larvae [42]. This protocol also was applied to reduce animal usage. From the results, 4 days of administration of p-TSA statistically reduced the larval responses to the vibrational stimuli. This phenomenon was indicated by the significant decrease in the area under curve (AUC) values observed in the treated groups compared to the untreated group (Figure 6A). Moreover, based on the average total distance traveled during the vibrations, all of the exposed groups also displayed statistically lower values than the control group, indicating the fewer response of these larvae toward the stimuli (Figure 6B,C).

### 3.6. Respiratory Rate Assessment of p-TSA in Zebrafish

The altered locomotion level displayed by the treated larvae in both locomotion-based assays motivated us also to evaluate the respiratory effect of p-TSA on zebrafish larvae. Therefore, the respiratory assay was carried out after 1-day incubation with this compound. From the results, all treated groups consumed more oxygen compared to the control group, indicating the high respiratory rate of these groups (Figure 7A,B). Furthermore, the oxygen consumption level increased with the increase of p-TSA concentrations. Taken together, 1-day exposure of p-TSA in the tested dosages resulted in respiratory toxicity to zebrafish larvae.

### 3.7. Molecular Docking of p-TSA to Its Target Protein

Molecular docking of p-TSA was ranked based on the binding energy obtained against the selected protein receptors. p-TSA displayed binding energies with v-type proton ATPase subunit A (−6.6 kcal/mol), v-type proton ATPase subunit H (−6.3 kcal/mol), ATPase GET3 (−5.9 kcal/mol), Myosin XIX (−5.9 kcal/mol), Phospholipid-transporting ATPase (−5.7 kcal/mol), Calcium-transporting ATPase (−5.6 kcal/mol), Sodium potassium transporting ATPase subunit alpha (−5.6 kcal/mol), v-type proton ATPase subunit C (−5.2 kcal/mol), Sodium/potassium-transporting ATPase subunit-beta−1-interacting protein 1 (−5.1 kcal/mol), Fast Skeletal Muscle Troponin C (−4.8 kcal/mol), Sodium/potassium-transporting ATPase subunit Beta (−4.7 kcal/mol), and Cardiac Troponin C (−4.6 kcal/mol). The interaction of p-TSA with v-type proton ATPase subunit H, Calcium-transporting ATPase, sodium-potassium transporting ATPase subunit Alpha, v-type proton ATPase subunit C, and Sodium/potassium-transporting ATPase subunit Beta is governed mostly by hydrophilic bond. For instance, p-TSA binds with the key amino acid residue of Calcium-transporting ATPase via Conventional Hydrogen Bond with Glu1000 and Arg918, Carbon Hydrogen Bond with Lys858, Pi-Cation with Lys858, Pi-Anion with Glu909, and Unfavorable Donor-Donor with Ile997 (Figure A1). Meanwhile, the interaction of p-TSA on the v-type proton ATPase subunit A, ATPase GET3, myosin XIX, phospholipid-transporting ATPase, sodium/potassium-transporting ATPase subunit-beta-1-interacting protein 1, fast skeletal muscle troponin C, Cardiac troponin C is governed by hydrophobic interactions. As shown in Figure A1, p-TSA binds with the key amino acid residue of v-type proton ATPase subunit A via Conventional Hydrogen Bond with Leu800, Pi-Sigma with Phe808, Pi-Pi Stacked with Phe808, Pi-Pi T-shaped with Phe451, and Pi-Alkyl with Ile458.

### 3.8. Cardiovascular Development-Related Genes Expression

Although no significant cardiac performance alteration was observed in the treated groups, abnormalities were shown in their locomotion with a minor alteration in their vascular performance. Therefore, a deeper investigation was conducted to help us in elucidating these phenomena by evaluating the expression level of several heart marker genes by using qRT-PCR. From the results, the expression levels of some heart developmental genes, vmhc and gata4, were downregulated, especially in the low concentration groups (Figure 8D,F). In addition, the expression level of hbbe1, one of the cardiovascular development-related genes, was also decreased after being treated with p-TSA, especially in the high concentration groups (Figure 8G). Meanwhile, no difference was observed in the expression level of other tested cardiovascular development-related genes, which were *amhc*, *myh6*, *vegfaa*, *gata1*, *hbbe2*, *hbae1*, and *tbx5* (Figure 8A–C,E,H–J). Taken together, acute exposure to p-TSA in certain concentrations only slightly altered some cardiovascular development-related genes in zebrafish larvae.

## 4. Discussion

In the present study, we evaluated the p-TSA toxicities toward zebrafish larvae by conducting several assays: photomotor response, vibrational startle response, oxygen consumption, and cardiotoxicity tests. To the best of our knowledge, this is the first study to evaluate the toxicity of this compound in fish, specifically zebrafish. Based on the results, acute exposure to p-TSA resulted in the hyperactivity-like behavior of zebrafish larvae in the photomotor assay and increased respiratory rate. However, no abnormalities were found in their morphology, specifically in their body and eyes axis length (Figure A2). Interestingly, a more extended exposure period caused a reduction in their response while stimulated with vibration stimuli during the vibration startle response assay. Meanwhile, no significant cardiac performance alterations were displayed in the tested larvae after 24 h of exposure to p-TSA. Taken together, even though the exact mechanism of its toxicity is still unaddressed in the current report, the toxic effects of p-TSA at tested concentrations on zebrafish are confirmed in this experiment.

Here, acute exposure to p-TSA, especially in high doses, caused high locomotor activity in zebrafish larvae. Interestingly, a similar outcome was also demonstrated in a prior prenatal study. After exposure to toluene, one of the compounds where p-TSA is derived, children up to 3 years old showed developmental delays, language impairment, hyperactivity, cerebellar dysfunction, and postnatal growth retardation [53,54,55]. In rats, acute administration of toluene resulted in reduced sleeping and increased locomotor activity during light periods, similar to the current results, and their brain monoamine metabolisms in circadian rhythms were disturbed [56,57]. Together with the current results, all of these findings suggest that p-TSA might share a similar mechanism with toluene in altering the behaviors of an individual. Toluene is a well-known solvent that can depress the central nervous system and irritate the respiratory tract [58]. Because of its high solubility to lipid tissues, it is easily absorbed into the central nervous system (CNS) and damages this system [59,60]. Recent research suggests that relatively low doses of inhaled toluene can adversely affect brain neurochemistry [61]. Numerous studies in various animals, including rats and pigeons, also demonstrated the effects of toluene on specific functions of the CNS, such as stimulus discrimination, time discrimination, and short-term memory, primarily cognitive functions of the CNS [62,63,64,65]. Interestingly, while their locomotion was increased during the photomotor test, suppressed locomotor activities were observed in the vibrational startle response assay after longer exposure. As we expected, this phenomenon was also observed in a prior study that found a decrease in the zebrafish larval spontaneous swimming activity after chronic exposure to three sulfonamides (sulfamethoxazole, sulfadiazine, and sulfadimidine). They hypothesized that this phenomenon occurred because sulfonamides may act as gamma-aminobutyric acid (GABA) agonists, causing acute neurological effects and damaging the coordination ability [30,66]. The difference in the outcome from these behavior tests might relate to the biphasic effects of toluene that have been reported in several prior studies. Taken together, we hypothesized that these effects of toluene on activity levels occurred due to the difference in exposure times in the vibration assay and the photomotor test that might explain the changes in the response of zebrafish to the stimuli in both assays. Furthermore, from the present docking results, p-TSA showed a binding affinity with Fast Skeletal Muscle Troponin C (TnC), a significant factor in developing and organizing thick and thin filaments. Based on these results, we also speculated that the less response of larval zebrafish recorded during the startle reflex experiment might be due to this binding since the previous study found that treating zebrafish larvae with BTS (N-benzyl-p-toluene sulphonamide), a myosin ATPase inhibitor, induced paralysis in fish larvae although they appeared morphologically normal except for the lack of muscle contraction [67]. However, molecular interactions provided by docking are not sufficient evidence to explain the mechanism involved in observed neurotoxicity, altered locomotion, and less response of zebrafish to p-TSA exposure. Nevertheless, the present study provides strong evidence that indicates p-TSA neurotoxicity toward zebrafish larvae after both acute or chronic exposures, and follow-up studies of the current docking results with genetic studies using p-TSA as an inhibiting agent are required to elucidate the effects of this compound on the expression of these proteins or receptors.

Next, acute exposure to p-TSA also resulted in statistically higher oxygen consumption rates in treated groups than in the untreated group. This phenomenon might cause a hypoxia condition that eventually leads to the death of the larvae. A similar result was mentioned in a prior study which found that toluene caused linear increases in the breathing rate of pink salmon fish (*Oncorhynchus gorbuscha*) with increasing exposure concentration, which is in line with the present study, indicating an increase in energy demands. This response was required for the metabolism of polycyclic aromatic hydrocarbons process since a variety of salmonids has been identified to possess an ability to metabolize oil-derived hydrocarbons [68,69,70]. In a study on aquatic stages of the mosquito (*Aedes aegypti* (L.), together with benzene or xylenes, toluene in sublethal doses produced a significant increase in oxygen consumption in treated fed-larvae. Later, they speculated that the mechanisms involved in mediating such respiratory responses are related to the synergistic effects of benzene and toluene that may be related to cell permeability since one of them may alter cell membranes and make them more permeable for the other [71]. Nevertheless, this result provides more evidence that shows the similarity of p-TSA and toluene toxicities in some specific parts of zebrafish larvae. In addition, this result might also result from the high locomotion observed in the photomotor test since oxygen consumption generally increases with swimming speed, as mentioned in previous studies in various fish and other organisms, including *Paramecium* [72,73,74,75]. In fish, this phenomenon might be accompanied by an increase in ventilation rate, which is intriguing to verify in future studies. Furthermore, it is also interesting to evaluate whether chronic exposure to PTS could result in more severe effects, such as oxygen displacement and respiratory paralysis, or not.

After acute exposure, although p-TSA was found to decrease the heart rate of zebrafish larvae, no significant alteration in cardiac performance was observed since high stroke volumes, were found in the treated groups, especially in the highest dose group. Generally, the lower heart rate is compensated by the higher stroke volume, resulting in the resting cardiac output being similar to that of a sedentary individual [76]. This phenomenon might also explain the significant increase in the blood flow speed of the treated zebrafish larvae, especially in the higher concentrations, since this parameter was regulated by several factors, including the pumping capability of the heart muscle, vasoconstriction, and vasodilation of smooth muscle of blood vessel [77,78]. This phenomenon might be explained by the slight decrements in the expression level of *gata4*, a gene that plays a major role during heart development [79]. Therefore, p-TSA in the given concentration disturbed larval heart development, resulting in a minor cardiac physiology alteration that might indicated by the abnormal stroke volume, which eventually was compensated by the adjustment of the blood flow as observed in the treated groups [80,81]. In addition, the irregular expression level of *hbbe1*, a hemoglobin marker gene, might also be related to this outcome since this irregularity could cause a premature return of caudal vein blood flow and impair the organization of mesencephalic vein and other brain blood vessels [82,83]. This effect on their brain might also elucidate the behavior alterations displayed by the exposed larvae in all of the behavior tests. Interestingly, while the results from other assays indicated the similarity of p-TSA and toluene effects toward zebrafish larvae, the current cardiac performance assay shows a contrasting result. Several studies provided substantial information on the hazardous effects of acute toluene on the cardiovascular system, indicating the heart as a sensitive target to toluene, with ventricular sensitization and arrhythmic effects along with tachycardia or bradycardia as the results [84,85,86,87]. In addition, many studies had already investigated that the changes in heart rates and ECG were the most reported cases of the toluene effect on cardiac function [88,89,90]. Moreover, cardiac toxicity occurs in some cases throughout its direct or indirect cardiotoxic effects, even in acute exposure with relatively low concentrations [91]. The current results also contradicted another finding from another study in sulfonamide. In their study, the heartbeat rate of zebrafish larvae showed an increasing trend with exposure to increasing concentrations of sulfonamides, although their incubation period was longer than our study [30]. However, one must keep in mind that besides *gata4*, acute exposure to p-TSA also caused abnormalities in the expression of another tested cardiac gene, which was *vmhc*. This gene is initially expressed in the anterolateral mesoderm and subsequently, its expression is restricted to the cardiac ventricle [92]. However, the slight alterations in its expression level observed in the study seemed not sufficient to cause noticeable cardiotoxicity. This might have occurred since no abnormalities were found in the expression levels of other tested cardiac development marker genes, thus; the deficiency might have been overcome by other genes. In addition, *vmhc* expression in the atrium is mediated by the *nxk2.5* gene [93]. Therefore, we hypothesized that p-TSA does not affect this gene so the abnormality in the *vmhc* expression could still be controlled, which further study is required to verify this hypothesis. Taken together, these results might indicate that p-TSA in the tested concentrations does not cause any cardiotoxicity when acutely administered to zebrafish larvae. However, there is still a possibility that after chronic exposure to p-TSA, tachyarrhythmia, a classical manifestation of toluene cardiotoxicity, will occur, which might be due to sensitization of the myocardium to the potential arrhythmogenic effect of endogenous catecholamine that is occasionally resulting in fatality [84,86,94].

In summary, despite its low toxicity level on zebrafish larvae, our results showed that p-TSA in relatively low concentrations resulted in behavior alterations, supported by the low respiratory rates in zebrafish larvae that, interestingly, are somewhat similar to the results from many previous studies on toluene (Figure 9). In silico investigation results revealed the molecular interaction between p-TSA with V-ATPase and TnC, wherein disturbances in their activities can impact cell proliferation, migration, or survival, and block muscle contraction that can lead to behavioral abnormalities. In this study, p-TSA did not cause any significant cardiotoxicity in zebrafish, as indicated by the normal expression levels of several major cardiac development marker genes (Figure 9). However, it exerted toluene-like effects in other parameters. Therefore, the binding affinities of this compound with Na^+^–K^+^-ATPase might only cause a minor effect on zebrafish larval cardiac activity, at least after acute exposure. Finally, the present study demonstrated p-TSA neurotoxicity toward zebrafish larvae, and future studies are required to investigate its effects after chronic exposure and provide an exact mechanism of its neurotoxicity.

## Figures and Tables

**Figure 1 biomolecules-12-01103-f001:**
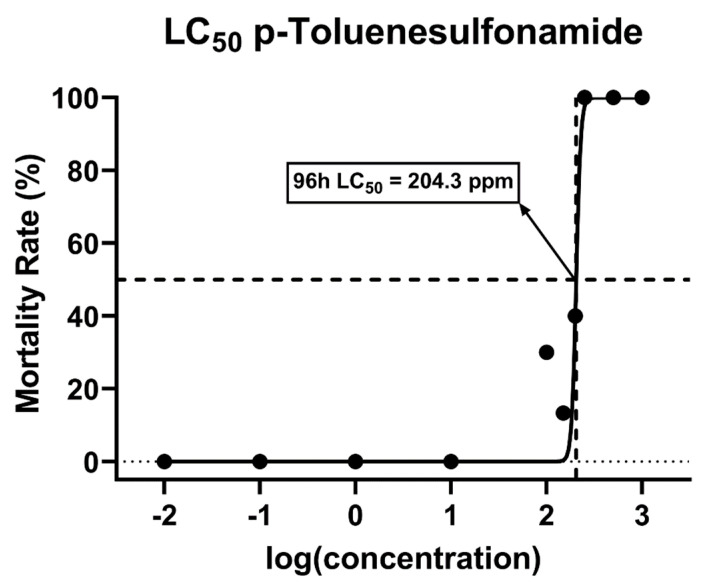
The mortality rate of zebrafish larvae after 96 h of exposure to p-Toluene Sulfonamide (p-TSA) to determine the median lethal dose (LC_50_).

**Figure 2 biomolecules-12-01103-f002:**
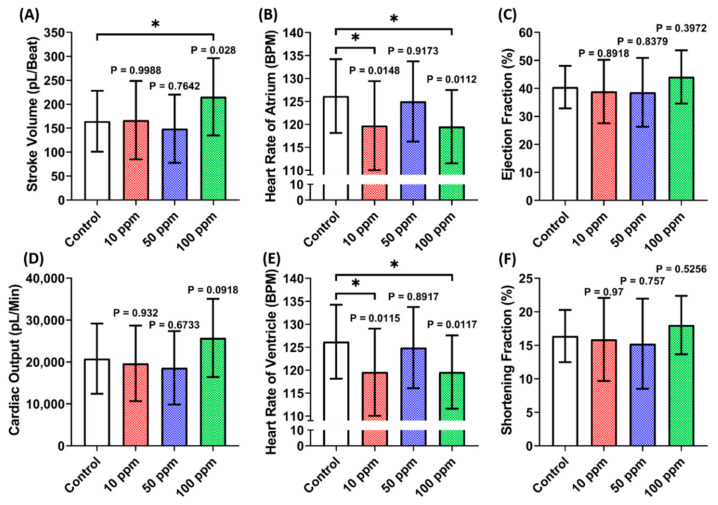
Cardiac physiology parameter endpoints ((**A**) Stroke volume, (**B**) heart rate of atrium, (**C**) ejection fraction, (**D**) cardiac output, (**E**) heart rate of ventricle, and (**F**) shortening fraction) of zebrafish larvae at 72 hpf after 24 h incubation in 10, 50, and 100 ppm of p-Toluene Sulfonamide (p-TSA). The statistical difference was calculated using Ordinary One-Way ANOVA with Dunnet multiple comparison test. The data are expressed as mean with SD (*n* = 29; * *p* < 0.05).

**Figure 3 biomolecules-12-01103-f003:**
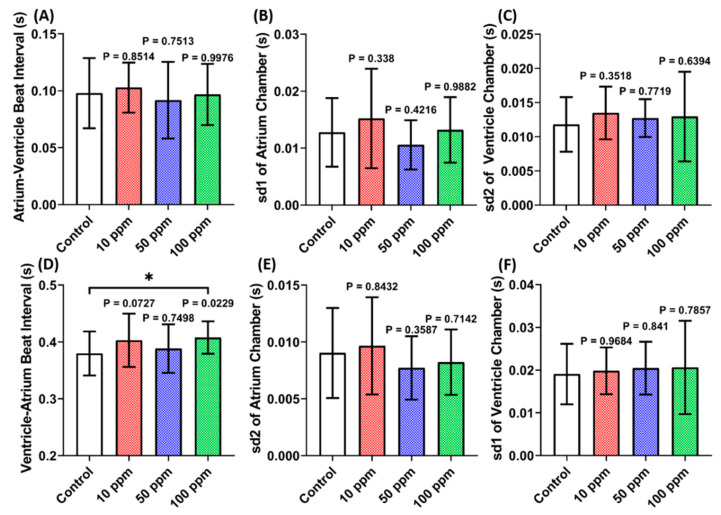
Cardiac rhythm parameter endpoints ((**A**) Atrium-ventricle beat interval, (**B**) sd1 of atrium chamber, (**C**) sd2 of ventricle chamber, (**D**) ventricle-atrium beat interval, (**E**) sd2 of atrium chamber, and (**F**) sd1 of ventricle chamber)of zebrafish larvae at 72 hpf after 24 h incubation in 10, 50, and 100 ppm of p-Toluene Sulfonamide (p-TSA). The statistical difference was calculated using Ordinary One-Way ANOVA with Dunnet multiple comparison test. The data are expressed as mean with SD (*n* = 29; * *p* < 0.05).

**Figure 4 biomolecules-12-01103-f004:**
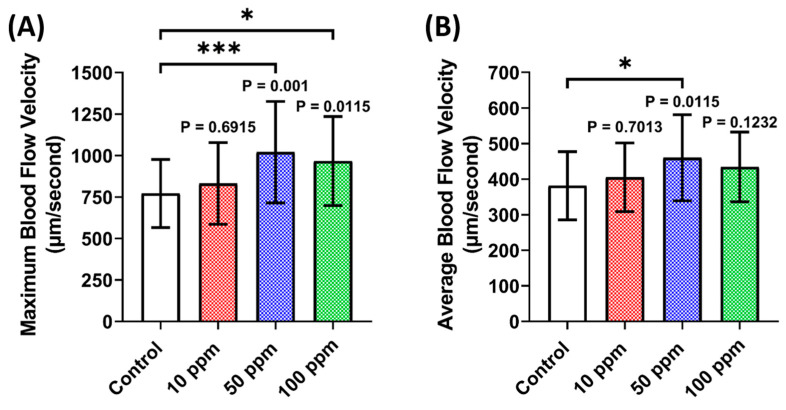
Vascular performance parameter endpoints ((**A**) Maximum blood flow velocity and (**B**) average blood flow velocity) of zebrafish larvae after 24 h incubation in 10, 50, and 100 ppm of p-Toluene Sulfonamide (p-TSA). The statistical difference was calculated using Ordinary One-Way ANOVA with Dunnet multiple comparison test. The data are shown as mean with SD (*n* = 30, except 50 ppm group (*n* = 29); * *p* < 0.05, *** *p* < 0.001).

**Figure 5 biomolecules-12-01103-f005:**
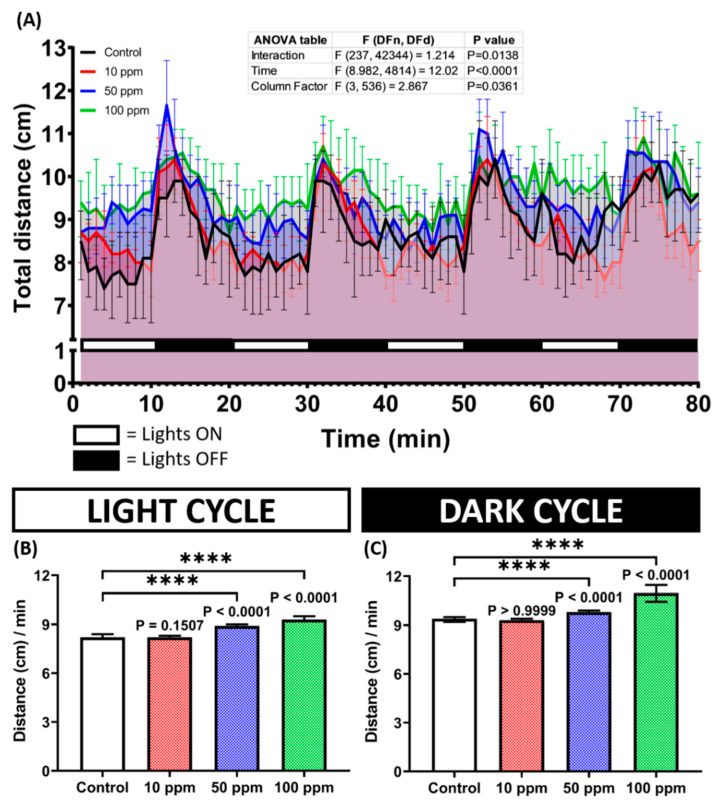
(**A**) Total distance traveled per minute by 128 hpf zebrafish larvae after 1-day exposure of 0 (control), 10 ppm, 50 ppm, and 100 ppm of p-Toluene Sulfonamide (p-TSA) during both light and dark cycles. The data were analyzed by a two-way ANOVA test with Geisser-Greenhouse correction, continued with Dunnett’s multiple comparisons test. (**B**,**C**) Comparison of total distance traveled by the larvae in light and dark cycles, respectively. The data were analyzed using the Kruskal–Wallis test, followed by Dunn’s multiple comparisons test. All data are expressed in median with 95% CI (*n* = 135 for control and 10 ppm groups, *n* = 134 for 50 ppm group, *n* = 136 for 100 ppm group; **** *p* < 0.0001).

**Figure 6 biomolecules-12-01103-f006:**
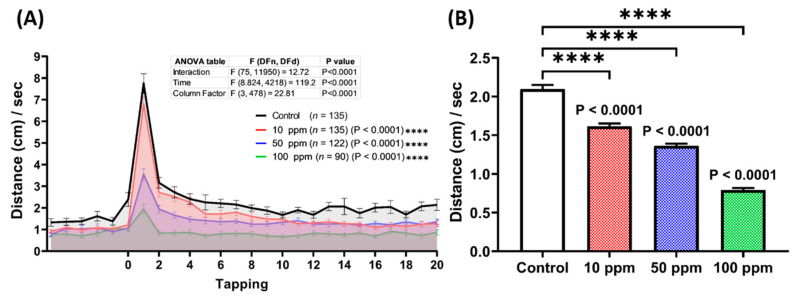
(**A**) Total distance traveled per second by 192 hpf zebrafish larvae after 4-day exposure of 0 (control), 10, 50, and 100 ppm of p-Toluene Sulfonamide (p-TSA) during the vibrational startle response assay. The data were analyzed using a two-way ANOVA test with Geisser-Greenhouse correction, followed by Dunnett’s multiple comparisons test. (**B**) A comparison of the total distance traveled by the tested zebrafish larvae during the occurrence of the tapping stimuli. The data were analyzed using the Kruskal–Wallis test, continued with Dunn’s multiple comparisons test. All data are expressed in median with 95% CI (*n* = 135 for control and 10 ppm groups, *n* = 122 for 50 ppm group, *n* = 90 for 100 ppm group; **** *p* < 0.0001).

**Figure 7 biomolecules-12-01103-f007:**
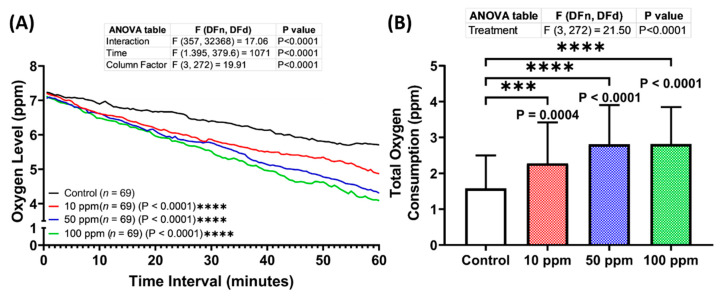
(**A**) Oxygen consumption level per minute by 96 hpf zebrafish larvae after 1-day exposure of 0 (control), 10, 50, and 100 ppm of p-Toluene Sulfonamide (p-TSA) during the respiratory rate assay. The data were analyzed using a two-way ANOVA test with Geisser-Greenhouse correction, followed by Dunnett’s multiple comparisons test. (**B**) Comparison of total oxygen consumption of the tested zebrafish larvae. The data were analyzed using one-way ANOVA, continued with Dunnett’s multiple comparisons test. All data expressed in the median with 95% CI (*n* = 69; *** *p* < 0.001, **** *p* < 0.0001).

**Figure 8 biomolecules-12-01103-f008:**
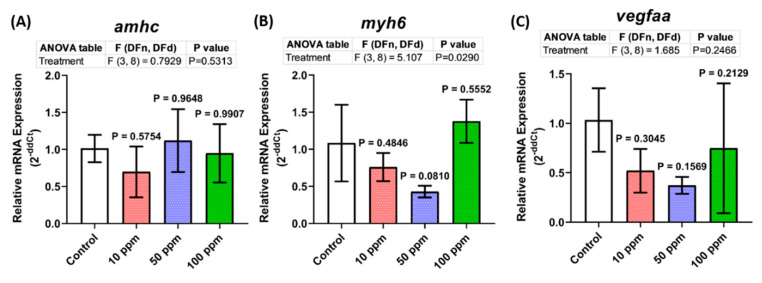
The expression pattern of cardiovascular development-related genes ((**A**) *amhc*, (**B**) *myh6*, (**C**) *vegfaa*, (**D**) *vmhc*, (**E**) *gata1*, (**F**) *gata4*, (**G**) *hbbe1*, (**H**) *hbbe2*, (**I**) *hbae1*, and (**J**) *tbx5*) in 5 dpf zebrafish larvae after 1 day exposure of p-Toluene Sulfonamide (p-TSA). The data were analyzed by using one-way ANOVA test, followed with Dunnett’s multiple comparisons test, and are presented as mean with SD (*n* = 3 groups with a total of 210 zebrafish larvae; * *p* < 0.05,** *p* < 0.01).

**Figure 9 biomolecules-12-01103-f009:**
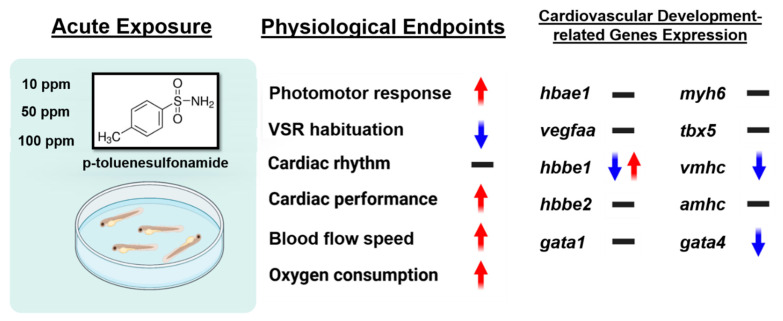
Summary of the present study demonstrated various alterations that occurred in zebrafish larvae after p-Toluene Sulfonamide (p-TSA) exposure (↑: upregulated, ↓: downregulated, -: no significant change).

**Table 1 biomolecules-12-01103-t001:** Dimension of the grid box that locates and encloses the binding site.

Proteins (UniProt)	Grid Box Parameters (x, y, z Coordinates)
ATPase GET3 (Q6IQE5)	Center: x = −5.014, y = −5.685, z = 2.564 Dimensions (Å): x = 64.180, y = 73.503, z = 89.405
Calcium-transporting ATPase (A0A286Y8X6)	Center: x = 11.827, y = 14.899, z = 3.753 Dimensions (Å): x = 120.170, y = 138.445, z = 136.836
Cardiac troponin C (Q800V7)	Center: x = 11.827, y = 14.899, z = 3.753 Dimensions (Å): x = 120.170, y = 138.445, z = 136.836
Fast skeletal muscle troponin C (Q918U8)	Center: x = −6.144, y = 1.479, z = 0.281 Dimensions (Å): x = 53.230, y = 36.770, z = 61.960
Myosin XIX (A0A0R4IEQ7)	Center: x = 3.520, y = 0.499, z = −6.061 Dimensions (Å): x= 116.386, y = 76.255, z= 82.202
Phospholipid-transporting ATPase (A0A0R4IC01)	Center: x = 0.368, y = 3.647, z = −0.397 Dimensions (Å): x = 117.422, y = 87.093, z = 112.228
Sodium potassium transporting ATPase subunit alpha (A0A0R4IJ10)	Center: x = −10.402, y = −0.074, z = −5.433 Dimensions (Å): x = 103.253, y = 76.181, z = 109.807
Sodium/potassium-transporting ATPase subunit Beta (Q9DGL3)	Center: x = −7.726, y = 10.487, z = −22.818 Dimensions (Å): x = 69.418, y = 70.062, z= 111.353
Sodium/potassium-transporting ATPase subunit-beta−1-interacting protein 1 (Q6PHL4)	Center: x = 19.534, y = 2.194, z = −12.666 Dimensions (Å): x = 96.393, y = 59.773, z = 85.675
v-type proton ATPase subunit A (A0A286Y8R2)	Center: x = −34.780, y = −3.090, z = −6.351 Dimensions (Å): x = 169.683, y = 84.065, z = 17.886
v-type proton ATPase subunit C (F1QPC9)	Center: x = 1.970, y = −4.212, z = −5.107 Dimensions (Å): x = 92.620, y = 73.399, z = 107.690
v-type proton ATPase subunit H (B0R0V8)	Center: x = −15.095, y = −3.567, z = 3.838 Dimensions (Å): x = 103.359, y = 59.900, z = 88.660

## Data Availability

The data presented in this study are available directly from the corresponding authors.

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
