# Peer review of "Toxicity Assessment of an Anti-Cancer Drug of p-Toluene Sulfonamide in Zebrafish Larvae Based on Cardiovascular and Locomotion Activities"

_biomolecules, 2022, doi:10.3390/biom12081103_

Round 1

Reviewer 1 Report

The manuscript by YouWah et al. studied toxic effects of p-toluene sulfonamide in zebrafish larvae.  Behavior effects were most prominent.  Other effects were minor.  The authors also performed modeling studies, producing candidate molecular interactions.  The studies were well-performed, and the manuscript was clearly written.  My biggest concern was that the authors speculated far too much about the implications of the molecular docking studies.  There was no experimental validation of the binding, and there was no evidence that the interactions caused activity changes.  There is no need to discuss the implications at length.  The findings would only be useful for hypothesis development.  Otherwise, the paper appears to be a strong contribution to the field.

Major concerns

1. The authors are not examining embryos.  The zebrafish are larvae at the stages examined.

2. Line 275: One does not verify an hypothesis.  One can only falsify a hypothesis or not.  The word “verify” should be replaced with the word “test.”

3. The morphology data should be reported in the results, not the discussion.  Also, what were the effects on morphology with higher concentrations.  This may be instructive to the lethal consequences of exposure.

4. There are so many comparisons to toluene in the discussion.  The authors should compare effects of p-toluene sulfonamide to effects of toluene, or the authors should minimize the discussion of toluene effects.

5. Remove or extremely minimize speculation about the molecular docking from lines 390-435 and 490-520.

Minor concerns

1. Line 66-67: “its toxicity studies” is awkward.  Perhaps, “toxicity studies of p-TSA are minimal” would be better.

2. Line 69: remove one “of”.

3. There are spaced needed between words: Line 69, “reportedin”; Line 89, “presentstudy”; Line 103, “previousstudy”; Line 134, “anddark”; and Line 135, “Beforethe”.

4. Line 98: The second “100” should be “1,000”.

5. Line 386: There are 2 commas after and no space before “In the present study.”

Author Response

Comments and Suggestions for Authors

The manuscript by YouWah et al. studied toxic effects of p-toluene sulfonamide in zebrafish larvae.  Behavior effects were most prominent.  Other effects were minor.  The authors also performed modeling studies, producing candidate molecular interactions.  The studies were well-performed, and the manuscript was clearly written.  My biggest concern was that the authors speculated far too much about the implications of the molecular docking studies.  There was no experimental validation of the binding, and there was no evidence that the interactions caused activity changes.  There is no need to discuss the implications at length.  The findings would only be useful for hypothesis development.  Otherwise, the paper appears to be a strong contribution to the field.

Major concerns

  1. The authors are not examining embryos.  The zebrafish are larvae at the stages examined.

The authors agreed with the reviewer’s comment. It is true that in the present study, the toxicities of p-TSA were examined in the zebrafish in their larval stage, not the embryo stage. Therefore, all of the improper “embryos” terms were changed to “larvae” as the reviewer suggested.

  1. Line 275: One does not verify an hypothesis.  One can only falsify a hypothesis or not.  The word “verify” should be replaced with the word “test.”

Thank you for the constructive suggestion. The authors admitted that there was a misuse of the word since the hypothesis was not being verified, instead, it was being tested. Thus, the word was changed according to the reviewer’s suggestion.

  1. The morphology data should be reported in the results, not the discussion.  Also, what were the effects on morphology with higher concentrations.  This may be instructive to the lethal consequences of exposure.

The authors thanked the reviewer for the suggestion. As the reviewer suggested, the morphology data were added to the results section, and some information regarding this data was also included in the materials and methods part. Regarding the effects on morphology with higher concentration, unfortunately, they were not evaluated in the current study since here, the main aim of the current morphology assessment was to show that the p-TSA effects on behaviors were not caused by the abnormal morphology; instead, they might be directly affected the central nervous system. Therefore, that is the reason why the authors only used morphology endpoints that are closely related to the larval locomotion, such as body length, and put these data on supplementary materials.

  1. There are so many comparisons to toluene in the discussion.  The authors should compare effects of p-toluene sulfonamide to effects of toluene, or the authors should minimize the discussion of toluene effects.

Thank you for the suggestion. The authors were aware that in the present manuscript, there were so many toluene comparisons in the discussion part. Initially, these comparisons were included because the number of prior p-TSA studies is very scarce despite their high usability in many fields, especially the medical field. Therefore, the authors hoped that those discussions regarding the toluene’s effect might help in understanding the toxicities of p-TSA. However, the authors agreed with the reviewer, and considering the limited study of p-TSA effects, the authors decided to minimize the discussion of toluene effects as the reviewer suggested.

  1. Remove or extremely minimize speculation about the molecular docking from lines 390-435 and 490-520.

The authors agreed with the reviewer. It is true that there were too many speculations from the molecular docking results regarding the toxicity mechanism of p-TSA. Therefore, as the reviewer suggested, the authors had tried their best to extremely minimize some parts of the discussion and entirely remove the unnecessary ones.

Minor concerns

  1. Line 66-67: “its toxicity studies” is awkward.  Perhaps, “toxicity studies of p-TSA are minimal” would be better.

Thank you for the suggestion. The authors agreed with the reviewer and thus, the sentence was revised according to the reviewer’s suggestion.

  1. Line 69: remove one “of”.

The authors appreciated the detailed check by the reviewer. The change was made as the reviewer suggested.

  1. There are spaced needed between words: Line 69, “reportedin”; Line 89, “presentstudy”; Line 103, “previousstudy”; Line 134, “anddark”; and Line 135, “Beforethe”.

The authors thanked the reviewer for the corrections. The mistyping might have occurred during the file conversion in adjusting the format of the journal. Therefore, the mistyping had been corrected according to the reviewer’s suggestion.

  1. Line 98: The second “100” should be “1,000”.

Thank you for pointing out the mistake. The information regarding the used concentrations in the current acute toxicity test was corrected as the reviewer suggested.  

  1. Line 386: There are 2 commas after and no space before “In the present study.”

The authors appreciated the comment. One of the commas was removed from the sentence according to the reviewer’s suggestion.

Reviewer 2 Report

The manuscript by Young et al. describes the toxic effects of p-TSA, a novel small antineoplastic molecule and determine the docking effects of p-TSA to elucidate the mechanisms underlying its toxicity. The manuscript is well written and the science is sound. Below are some suggestions for improvement: 

1. The authors use embryo mortality rate at the 96-h timepoint to determine the acute toxicity of p-TSA to claim it is relatively non-toxic. Were any other morphogenic abnormalities i.e. delayed hatching, abnormal larvae etc. observed? 

2. The doses used in the study are much higher than the doses used in previous studies cited in the paper 'Recently, the toxic effects of of sulfonamide has been reported in zebrafish on the parameters of behavior and reproduction, sulfonamide comprising sulfadimidine, sulfamethoxazole, and sulfadiazine at a concentration range of 1 µg/L to 10 mg/L caused a decrease in swimming activity and an increase in heart rate of zebrafish [30].' The authors should provide a justification for the doses used in the manuscript. 

3. The authors claim that the increased stroke volume compensates for heart rate to maintain cardiac output. Is there an explanation why this effect is only seen at the 50 ppm dose? 

4. In addition to cardiac function, western blot or qPCR must be performed to confirm that cardiac markers are not altered by p-TSA and thus is non-cardiotoxic. 

5. The authors may consider including information about the tissue-specific expression of the proteins listed as docking targets of p-TSA

6. Lines 396-410 in the Discussion require references.

Author Response

Comments and Suggestions for Authors

The manuscript by Young et al. describes the toxic effects of p-TSA, a novel small antineoplastic molecule and determine the docking effects of p-TSA to elucidate the mechanisms underlying its toxicity. The manuscript is well written and the science is sound. Below are some suggestions for improvement: 

  1. The authors use embryo mortality rate at the 96-h timepoint to determine the acute toxicity of p-TSA to claim it is relatively non-toxic. Were any other morphogenic abnormalities i.e. delayed hatching, abnormal larvae etc. observed? 

The authors appreciated the questions from the reviewer. Unfortunately, the other morphogenic abnormality endpoints like the reviewer mentioned were not evaluated in the present study, since, the main purpose of the present embryo mortality test was to obtain the reference for the concentrations used in the following tests. Therefore, the mortality rate was only the parameter assessed in the test and later, by comparing the obtained lethality results with some of the reliable rating scales, p-TSA is categorized as a practically nontoxic compound to aquatic organisms.

  1. The doses used in the study are much higher than the doses used in previous studies cited in the paper 'Recently, the toxic effects of of sulfonamide has been reported in zebrafish on the parameters of behavior and reproduction, sulfonamide comprising sulfadimidine, sulfamethoxazole, and sulfadiazine at a concentration range of 1 µg/L to 10 mg/L caused a decrease in swimming activity and an increase in heart rate of zebrafish [30].' The authors should provide a justification for the doses used in the manuscript. 

Thank you for the comment. As the aim of the paper was to assess the possible risk of toxicity caused by p-TSA, the authors believed that it would be better to study its toxicity in the concentrations that are based on other results of p-TSA as well, instead of other compounds, such as its derivatives. Based on this consideration, the selected concentrations for all of the tests were referred to the LC50 value shown in this study, which was the sub-lethal concentrations, as mentioned above. In addition, the authors were also aware of the importance of the rationale for the current concentration selection. Therefore, to improve the reader's comprehension, some information regarding the used concentrations was added to the manuscript, specifically in the Materials and Methods section.

  1. The authors claim that the increased stroke volume compensates for heart rate to maintain cardiac output. Is there an explanation why this effect is only seen at the 50 ppm dose? 

The authors appreciated the reviewer’s question. Actually, the increased stroke volume that compensated for the heart rate was clearly shown in the 100-ppm group, instead of the 50-ppm group (Figure 2A). Unfortunately, due to the limited number of prior p-TSA studies, especially in the cardiovascular system, the authors could not certainly elucidate the exact reasons for this phenomenon. However, the authors only might connect this phenomenon with a previous finding, which reported the ability of p-TSA to replace ω-sidechain structures of various prostanoids, causing them to behave as Thromboxane A2 (TXA2) antagonists [1]. TXA2 is a type of thromboxane that is produced by activated platelets during hemostasis and has prothrombotic properties that might be related to the increased stroke volume. In addition, TXA2 is also a vasoconstrictor and its role is very important during tissue injury and inflammation [2,3]. Therefore, although the possibility of replacement of the ω-sidechain of various prostanoid structures by p-TSA was not presented, the authors hypothesized that this phenomenon might possibly occur when the p-TSA concentration was high and disturbed the cardiovascular system homeostasis. However, further studies have to be done to prove this hypothesis.

  1. Corey, E.; Wei-guo, S. Total synthesis of a potent thromboxane A2 antagonist. Tetrahedron letters 1990, 31, 3833-3836.
  2. Ding, X.; Murray, P.A. Cellular mechanisms of thromboxane A2-mediated contraction in pulmonary veins. American Journal of Physiology-Lung Cellular and Molecular Physiology 2005, 289, L825-L833.
  3. Smyth, E.M. Thromboxane and the thromboxane receptor in cardiovascular disease. Clinical lipidology 2010, 5, 209-219.
  4. In addition to cardiac function, western blot or qPCR must be performed to confirm that cardiac markers are not altered by p-TSA and thus is non-cardiotoxic. 

Thank you for the suggestion. The authors agreed with the reviewer, thus, the expression level of several cardiac development markers was measured by using qRT-PCR and the results were added to the manuscript. Overall, the expression level of most of the marker genes was unchanged after being treated with p-TSA. However, slight changes were observed in the vmhc expression level, which might be still not sufficient to cause pronounced cardiotoxicity in the zebrafish larvae. In addition, changes were also found in some of the blood marker genes, which might elucidate the high blood flow observed in the present study. The detailed information and discussion regarding this matter were added to the manuscript.

  1. The authors may consider including information about the tissue-specific expression of the proteins listed as docking targets of p-TSA

The 3D structure of docking target proteins was downloaded from AlphaFold protein structure database (https://alphafold.ebi.ac.uk/), and there are no gene expression-related information summarized in this database. We also try to find related gene expression information in ZFIN, however, too many isoforms were found and unable to make sure which one is corresponding to the docking targets. Therefore, we finally not including this information in the revised paper. 

  1. Lines 396-410 in the Discussion require references.

The authors thanked the reviewer for the suggestion. However, based on another reviewer’s suggestion, several sections of the discussion part, especially the molecular docking section, were unnecessary and could be removed from the manuscript. Therefore, the authors decided to remove several parts from the discussion section, including the one meant by the reviewer.

Round 2

Reviewer 2 Report

The authors have addressed all concerns in the revised manuscript. The manuscript is appropriate for publication